# Association between the retinal vascular network and retinal nerve fiber layer in the elderly: The Montrachet study

Louis Arnould[1,2,3,4][⊛]*, Martin Guillemin[1][⊛], Alassane Seydou[2,3,4], Pierre-Henry Gabrielle[1,4], Abderrahmane Bourredjem[2,3], Ryo Kawasaki[5], Christine Binquet[2,3], Alain M. Bron[1,4], Catherine Creuzot-Garcher[1,4]

1 Department of Ophthalmology, University Hospital, Dijon, France, 2 INSERM, CIC1432, Clinical Epidemiology Unit, Dijon, France, 3 Clinical Investigation Center, Clinical Epidemiology/Clinical Trials Unit, Dijon University Hospital, Dijon, France, 4 Centre des Sciences du Goût et de l'Alimentation, AgroSup Dijon, CNRS, INRAE, Université Bourgogne Franche-Comté, Dijon, France, 5 Department of Vision Informatics, Osaka University Graduate School of Medicine, Suita, Japan

⊛ These authors contributed equally to this work.
* louis.arnould@chu-dijon.fr

**Data Availability Statement:** Data are fully available if a request is made to the Montrachet Study Comittee an email has to be written to Prof

## Abstract

### Purpose

To investigate the association between the characteristics of the retinal vascular network in the elderly and retinal nerve fiber layer (RNFL) thickness in a population-based study.

### Methods

We conducted a population-based study, the Montrachet study (Maculopathy Optic Nerve, nuTRition neurovAsCular, and HEarT disease), in participants aged $\geq$ 75 years. RNFL thickness was assessed with spectral-domain optical coherence tomography (SD-OCT). Analysis of the retinal vascular network was performed by means of the Singapore "I" Vessel Assessment (SIVA) software based on fundus photography.

### Results

Data from 970 participants were suitable for analysis. Patients with optic neuropathy were excluded. In multivariable analysis, each standard deviation (SD) decrease in the caliber of the six largest arterioles and veins in zone B and the six largest arterioles and veins in zone C was associated with a decrease in global RNFL thickness ($\beta$ = −1.62 µm, $P$ = 0.001; $\beta$ = −2.39 µm, $P$ < 0.001; $\beta$ = −1.56 µm, $P$ = 0.002; and $\beta$ = −2.64 µm, $P$ < 0.001, respectively).

### Conclusions

Our study found that decreased retinal vessels caliber were associated with a decreased RNFL thickness in the elderly without optic neuropathy.

Tzourio Christophe christophe.tzourio@u-bordeaux.fr. There are legal restrictions on sharing the data because they are owned by the 3C study comittee. Each request has to be examined by the 3C Comittee with a full data request form. The authors had no special access privileges and other researchers would be able to access the data in the same manner as the authors.

**Funding:** This study was supported by an interregional grant (Programme Hospitalier de Recherche Clinique) and the Regional Council of Burgundy; by INRA, CNRS, Université de Bourgogne, Regional Council of Burgundy France (PARI Agrale 1), FEDER (European Funding for Regional Economic Development); and a French Government grant managed by the French National Research Agency (ANR) under the "Investissements d'Avenir" program, ANR-11-LABX-0021-01-LipSTIC Labex.

**Competing interests:** Disclosure: M. Guillemin, None; L. Arnould, None; A. Seydou, None; P H. Gabrielle, None; A. Bourredjem, None; R. Kawasaki, None; C. Binquet, None; A M. Bron, Aerie (C), Allergan (C), Baush and Lomb (C), Santen Pharmaceutical (C), Théa (C); C. Creuzot-Garcher, Allergan (C), Bayer (C), Horus (C), Novartis (C), Roche (C), Théa (C). This does not alter our adherence to PLOS ONE policies on sharing data and materials.

## Introduction

There is growing interest in the quantitative description of the retinal microvascular network due to the fact that numerous studies have shown associations between retinal vascular characteristics and both ocular and systemic diseases [1–6]. Retinal vessels are easy to study, thus providing a noninvasive view of the human microcirculation network using fundus photography. In order to describe the geometric characteristics of arterial and velnous components of the retinal network, specific software programs have been developed that can automatically analyze and describe its geometric characteristics. Retinal Analysis (RA) was the pioneer software but required significant user input and manual tracing of vessels [7]. Automated software, such as the Singapore "I" Vessel Assessment (SIVA) was later developed. SIVA software enables quantitative analysis of vascular morphometry and was shown to be accurate and reproducible in assessing multiple in vivo architectural changes in the retinal vascular network [8, 9].

Today, the assessment of retinal nerve fiber layer (RNFL) thickness is investigated with spectral-domain optical coherence tomography (SD-OCT). Age, refraction, and axial length are consistently associated with RNFL thickness in large studies [10–13]. The association between narrower retinal vessel caliber and RNFL thinning has been described in individuals with glaucoma [14–19]. However, few studies have investigated the relationship between RNFL thickness and retinal vessels in non-glaucomatous populations [20, 21].

The purpose of this study was to investigate the association between quantitative retinal vascular characteristics and RNFL thickness in eyes without optic neuropathy in a population-based study focused on the elderly.

## Methods

### Study design and population

The Montrachet (Maculopathy Optic Nerve and nuTRition neurovAsCular and HEarT disease) study was an ancillary study of the Three-City (3C) study that started in 1999 and was designed to examine the relationship between vascular risk and dementia [22]. The methodology of the Montrachet study and the participant and nonparticipant characteristics have already been described [23]. Briefly, elderly urban inhabitants from three French cities: Montpellier, Bordeaux and Dijon were recruited between October 2009 and March 2013. This study aimed to investigate potential associations between age-related eye diseases with neurological and cardiovascular pathologies in elderly participants. The study was approved by the 3C study ethics committee. Participants gave their written consent to the study. It followed the tenets of the Declaration of Helsinki and it was registered as 2009-A00448-49. We followed the STROBE (STrengthening the Reporting of OBservational studies in Epidemiology) statement according to the EQUATOR (Enhancing the QUAlity and Transparency Of health Research) guidelines [24].

Cardiovascular history as well as medication and lifestyle were collected for each participants of the 3C study based on self-declaration at baseline and every 2 years thereafter for 10 years. Cardiovascular events and ischemic stroke were summarized in a single variable: major adverse cardiovascular or cerebrovascular events (MACCE). Blood samples were collected after fasting. The 10-year risk of fatal cardiovascular disease was estimated by means of Montrachet baseline information using the method proposed in the Systematic COronary Risk Evaluation (Heart SCORE) project [25].

## SD-OCT RNFL thickness measurements

We performed SD-OCT (software version 5.4.7.0; Spectralis, Heidelberg Engineering Co., Heidelberg, Germany) of the optic disc after pupil dilation with tropicamide 0.5% (Théa, Clermont-Ferrand, France). Signal strength and the quality of segmentation were controlled by technicians based on their expertise. In the case of layer segmentation errors, manual correction was performed by the technician. RNFL thickness was measured around a 3.5-mm-diameter circle centered on the optic nerve head. The software calculated RNFL thickness and generated global and sector peripapillary RNFL thicknesses (nasal and temporal sectors).

## Fundus photographs and vessel analysis

The methodology used to extract retinal vessel characteristics with the SIVA assessment software has already been extensively described in our previous study [26]. Briefly, we used fundoscopic retinal photographs (45 degrees) centered on the optic disc using a fundus camera (TRC-NW6S; Topcon, Tokyo, Japan). Photographs were sent to an external reading center at Yamagata University, Japan. We only retained one eye for analysis. Computerized analysis of the retinal microvascularization was based on the analysis of venules and arterioles from the center of the optic disc and then to three successive zones corresponding to a circular zone from the edge of the optic nerve head to 0.5 (zone A), 1 (zone B), and 2 (zone C) optic disc diameters. Retinal microvascular biomarkers were measured in zones B and C. The main retinal vascular parameters analyzed were vessel caliber, fractal dimension, and vascular tortuosity. Fractal dimension is a mathematical measure that quantifies complex geometric patterns in objects that are self-similar in their scaling patterns [27]. Retinal vascular fractal dimension represents a global measure of retinal vascular network complexity and density that summarizes the branching pattern of the retinal vasculature [9]. A higher fractal dimension value indicates a more complex vascular branching pattern and a lower fractal dimension indicates a sparser vascular network. Retinal vascular tortuosity reflects the straightness of the vessels; this measure is dimensionless [28]. A higher tortuosity value indicates more curved retinal vessels.

## Exclusion criteria

Participants with glaucoma as defined by the International Society for Geographical and Epidemiological Ophthalmology were excluded [29]. Identified glaucomatous eyes were re-examined with Humphrey Swedish interactive thresholding algorithm 24–2 visual field (Carl Zeiss Meditec, Dublin, CA, USA) and a systematic gonioscopy. Optic disc photographs were interpreted by two trained ophthalmologists. In the case of discrepancy, the adjudication was made by a glaucoma specialist. To determine the three levels of evidence described by Foster et al., we used the 97.5th and 99.5th percentiles for the vertical cup-to-disk ratio found in our population-based study [29]. The 97.5th and 99.5th percentiles for the vertical cup-to-disk ratio and vertical cup-to-disk ratio asymmetry were 0.7, 0.2, and 0.8, 0.3, respectively. As previously reported, we also excluded participants presenting with epiretinal membrane from the analysis as they are known to modify vascular architecture [26]. Patients presenting with late age-related macular degeneration (AMD) and modification of the vascular architecture in zone B and C, attested by a trained grader, were not retained for the analysis. AMD was diagnosed on macular retinal photographs according to the modified international classification [30]. Participants with epiretinal membrane and AMD with modification of the vascular architecture were excluded because they present primary modifications of the retinal vascular profile without systemic involvement.

## Statistical analysis

Categorical variables as expressed as number (*n*, %) and continuous variables as mean ± standard deviation (SD) or median (interquartile range [Q1–Q3]) according to their distribution. Participants and non-participants were compared with the chi-square test or Fisher's exact test for categorical variables and the Student *t* test or ANOVA test for continuous variables where appropriate. Associations between axial length, Heart SCORE, and global RNFL were estimated with simple linear regression models. In order to estimate associations between retinal vascular parameters (caliber, fractal dimension, and vascular tortuosity) as independent variables and RNFL thickness (global, temporal, and nasal RNFL) as dependent variables, three individual multivariable linear regression models with one eye per individual as the unit of analysis were performed. All variables associated with global RNFL with *P* values less than 0.20 in bivariate analysis were included in the individual multivariable models except for Heart SCORE. First, models were systematically adjusted for age and sex. Second, final models were obtained after adjustment for age, sex, axial length, diabetes, and systemic hypertension. Associations were expressed as β (standard error, SE). For all analyses, the tests were two-tailed, and the results were considered significant when P-values were less than 0.05. Given the number of statistical tests in this study, we used the procedure of False Discovery Rate (FRD) correction to correct p values for multiple testing [31]. FDR Adjusted p values are presented in multivariable models for each independent variable. A separate analysis in participants without hypertension and diabetes was performed. Analyses were performed using SAS software (version 9.4; SAS Institute, Inc., Cary, NC, USA).

## Results

Of the 1153 subjects included in the Montrachet study, 970 eyes had suitable data for analysis (Fig 1). The characteristics of the study population and a comparison between participants and non-participants are summarized in the supporting information (S1 Table). The group of non-participants was significantly older (*P* < 0.001) and had a longer axial length (*P* < 0.001). Two participants with diabetes and diabetic retinopathy were kept in the analysis. The mean RNFL of the study participants was 92.08 ± 13.29 μm and the mean age was 82.06 ± 3.64 years old.

Table 1 presents the demographics, lifestyle, and clinical characteristics of participants according to global RNFL thickness. A significantly thinner global RNFL thickness was found in males (*P* = 0.025) and patients with diabetes (*P* = 0.012). Participants with longer axial length also presented a thinner global RNFL thickness, β = −3.59 μm (*P* < 0.001).

Table 2 presents the associations of retinal vascular parameters and global RNFL thickness. After adjustment for age and sex, a reduced caliber of arterioles and veins was significantly associated with a thinner global RNFL thickness (*P* < 0.001). Similar results were found with a reduced fractal dimension (*P* = 0.001 for total zone C, *P* = 0.005 for arteriole zone C, *P* < 0.001 for vein zone C) and tortuosity of vessels (*P* = 0.029). After further adjustment for age, sex, axial length, diabetes, and systemic hypertension, each SD decrease in the caliber of the six largest arterioles and veins in zone B and in the caliber of the six largest arterioles and veins in zone C was associated with a decrease in global RNFL thickness (β = −1.62 μm, *P* = 0.001; β = −2.39 μm, *P* < 0.001; β = −1.56 μm, *P* = 0.002; and β = −2.64 μm, *P* < 0.001, respectively). Finally, regarding fractal dimension, each SD decrease in the veins in zone C was associated with a 1.02-μm decrease in the global RNFL thickness (*P* = 0.028). In participants without hypertension and diabetes (n = 294), after adjustments for age, sex and axial length, each SD decrease in the caliber of the six largest arterioles and veins in zone B and in the caliber of the six largest arterioles and veins in zone C was associated with a decrease in global RNFL

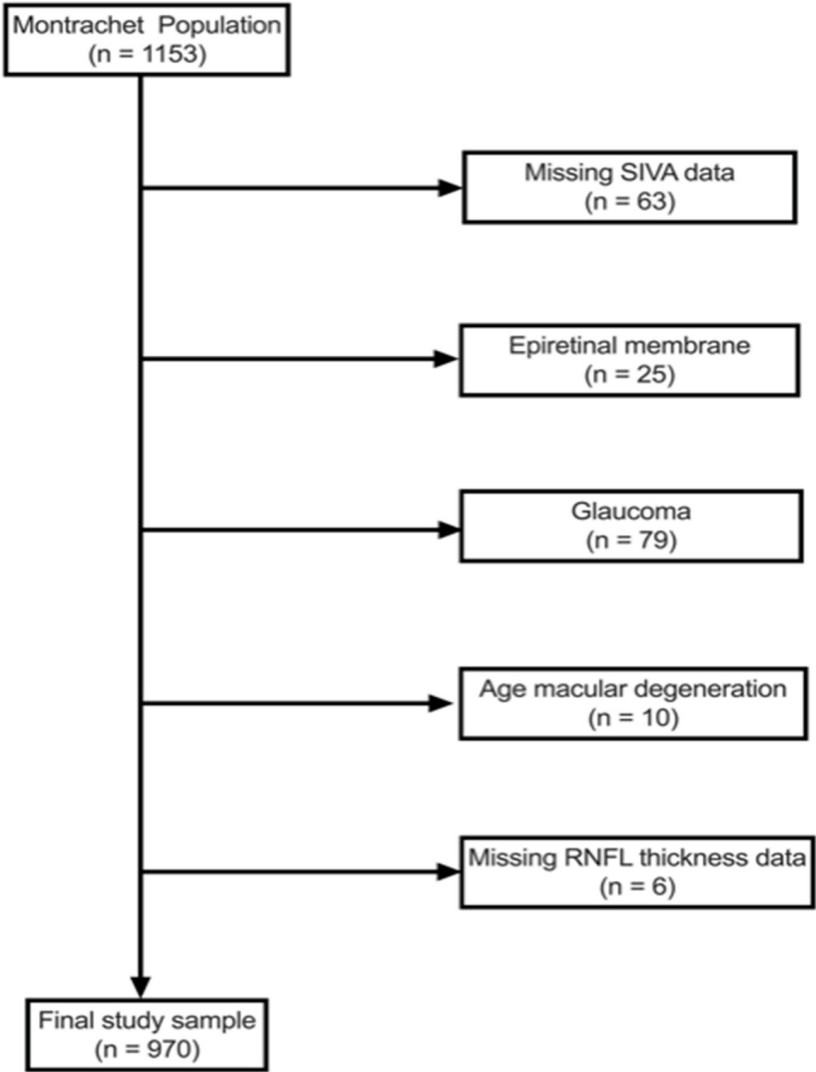

**Fig 1. Flow chart of the study.** *SIVA*, Singapore "I" Vessel Assessment; *RNFL*, retinal nerve fiber layer. Only patients with late age-related macular degeneration and modification of the vascular architecture in zone B and C were excluded.

thickness (β = −1.88 μm, P = 0.0371; β = −3.28 μm, P < 0.001; β = −2.02 μm, P < 0.001; and β = −3.27 μm, P < 0.001, respectively) (Table 3).

The analyses of the association between retinal vascular parameters and sectorial temporal and nasal RNFL are shown in S2 and S3 Tables.

## Discussion

In this population-based study of older adults, we found that thinner retinal vessels caliber and reduced fractal dimension of veins in zone C were associated with a thinner global RNFL thickness in non-glaucomatous participants. These results are in keeping with other published data. In a cross-sectional investigation of Asian participants, each SD decrease in retinal arteriolar caliber was associated with a 5.81 μm decrease in the mean global measured RNFL

**Table 1. Associations between retinal nerve fiber layer thickness and demographics, lifestyle, clinical characteristics of participants.**

| Baseline characteristics | Participants, n = 970 | Global RNFL, μm | P-value |
|---|---|---|---|
| Age, years | | | |
| <80 | 351 (36.18) | 92.25 ± 11.59 | 0.081 |
| 80–85 | 416 (42.89) | 92.80 ± 14.86 | |
| >85 | 203 (20.93) | 90.27 ± 12.49 | |
| Sex | | | |
| Male | 353 (36.39) | 90.81 ± 14.87 | 0.025 |
| Female | 617 (63.61) | 92.79 ± 12.25 | |
| Smoking status, self-declared | | | |
| Non-smokers | 512 (57.14) | 92.18 ± 12.25 | 0.894 |
| Smokers | 384 (42.86) | 92.30 ± 14.58 | |
| Diabetes, self-declared | | | |
| No | 761 (87.27) | 92.61 ± 13.04 | 0.012 |
| Yes | 111 (12.73) | 89.20 ± 15.05 | |
| Treatment for systemic hypertension | | | |
| No | 319 (34.52) | 92.51 ± 12.79 | 0.063 |
| Yes for < 10 years | 256 (27.71) | 93.27 ± 15.31 | |
| Yes for ≥ 10 years | 349 (37.77) | 90.51 ± 12.79 | |
| Cholesterol-lowering drug use | | | |
| No | 288 (30.90) | 92.24 ± 14.34 | 0.964 |
| Yes | 644 (69.10) | 92.19 ± 12.81 | |
| MACCE | | | |
| No | 811 (88.83) | 92.21 ± 13.10 | 0.243 |
| Yes | 102 (11.17) | 90.62 ± 12.10 | |
| Axial length, mm, per SD decrease, β (SE) | 23.21 ± 1.21 | -3.59 (0.37) | <0.001 |

The results are displayed as n (%) for categorical variables and mean ± standard deviation for continuous variables. RNFL, retinal nerve fiber layer; MACCE, major adverse cardiovascular or cerebrovascular events; SD, standard deviation; SE, standard error. Available data for each variable, smoking status n = 896; diabetes n = 872; treatment for systemic hypertension n = 924; cholesterol-lowering drug use n = 932; MACCE n = 913, axial length n = 798.

thickness ($P < 0.001$), and each SD decrease in retinal venular caliber was associated with an 8.37 μm decrease in mean global RNFL thickness ($P < 0.001$) [32]. In this study, participants' mean age was 56.8 years. In a study derived from the Singapore Chinese Eye Study, each SD decrease in the retinal arteriolar caliber was associated with a 1.60 μm decrease in global RNFL thickness ($P = 0.002$), and each SD decrease in venular caliber was associated with a 1.97 μm decrease in average RNFL thickness ($P < 0.001$) (mean age was 53.6 ± 6.7 years) [21]. Finally, this association was also confirmed in a middle-aged Caucasian population, where each 10 μm increase in the retinal arteriolar and venular caliber was associated with, respectively, a 5.58 μm and a 3.79 μm increase in mean global RNFL thickness ($P < 0.001$) (mean age was 47.3 ± 0.9 years) [20]. Moreover, in a subgroup analysis (participants without hypertension or diabetes), we found similar results. Thus we confirmed in this study the association between RNFL thickness and retinal vessels caliber in an elderly population.

When investigating the RNFL nasal sector, the results were similar to those found for the global analysis, whereas an opposite association was found when investigating the temporal sector. A reduction in the caliber of vessels and a reduction in fractal dimension were associated with increased RNFL thickness.

It remains unknown whether the relationship between retinal vessel narrowing and reduced RNFL thickness is causal. Although it is impossible to provide a definite answer to

**Table 2. Associations between retinal vascular parameters and global retinal nerve fiber layer thickness.**

| Retinal vascular parameters (per SD decrease) | Crude associations | | | Age, sex-adjusted | | Multivariable-adjusted* | | |
|---|---|---|---|---|---|---|---|---|
| | Mean ± SD | β (SE) | P-value | β (SE) | P-value | β (SE) | Unadjusted P-value | FDR Adjusted P-value |
| Caliber, μm | | | | | | | | |
| Six largest arterioles in zone B | 139.14 ± 13.80 | -3.16 (0.44) | <0.001 | -3.04 (0.43) | <0.001 | -1.62 (0.51) | 0.001 | 0.004 |
| Six largest veins in zone B | 194.39 ± 21.60 | -3.59 (0.43) | <0.001 | -3.52 (0.43) | <0.001 | -2.39 (0.49) | <0.001 | <0.001 |
| Six largest arterioles in zone C | 143.33 ± 14.54 | -3.00 (0.44) | <0.001 | -2.88 (0.44) | <0.001 | -1.56 (0.50) | 0.002 | 0.005 |
| Six largest veins in zone C | 202.12 ± 21.29 | -3.73 (0.43) | <0.001 | -3.67 (0.43) | <0.001 | -2.64 (0.49) | <0.001 | <0.001 |
| Fractal dimension | | | | | | | | |
| Total zone C | 1.38 ± 0.07 | -1.47 (0.42) | <0.001 | -1.35 (0.42) | 0.001 | -0.75 (0.47) | 0.106 | 0.159 |
| Arterioles zone C | 1.19 ± 0.07 | -1.26 (0.41) | 0.002 | -1.17 (0.41) | 0.005 | -0.59 (0.45) | 0.194 | 0.249 |
| Veins zone C | 1.16 ± 0.06 | -1.51 (0.42) | <0.001 | -1.43 (0.42) | <0.001 | -1.02 (0.46) | 0.028 | 0.042 |
| Vascular tortuosity | | | | | | | | |
| Simple tortuosity, vessels, × $10^4$ | 1.09 ± 0.02 | -2.15 (0.94) | 0.022 | -2.05 (0.93) | 0.029 | -0.94 (1.02) | 0.357 | 0.401 |
| Curvature tortuosity, vessels, × $10^4$ | 0.74 ± 0.17 | -0.75 (0.41) | 0.068 | -0.68 (0.41) | 0.099 | -0.30 (0.44) | 0.487 | 0.487 |

SD, standard deviation; SE, standard error; FDR, false discovery rate. 172 observations were deleted due to missing axial length variable.

*Adjusted for age, sex, axial length, diabetes, and systemic hypertension.

this question, it is plausible that these changes are related to vascular autoregulation. Higher levels of oxidative factors have been found to lead to vasodilatation and elevated blood flow and could influence retinal vessel diameter [33]. It was also postulated that reduced venular caliber could lead to cytotoxic and vasogenic edema. This congestion would then cause arteriolar constriction and finally lead to early RNFL thinning. This vascular involvement seems to be reinforced, since vascular and neurovascular diseases such as stroke and dementia were shown to be associated with reduced RNFL thickness [34].

Fractal dimension was also investigated in our study. This parameter reflects the complexity of the retinal microvascular tree as a geometric pattern including the degree of branching [27]. Optimal retinal vasculature with a higher fractal dimension is associated with greater efficiency in blood circulation [10, 35]. It has been reported to be associated with age, blood pressure, refractive errors, and lens opacity [3, 36]. Low fractal dimension was associated with cardiovascular mortality risk and advanced diabetic retinopathy [26, 37]. In our elderly population, a lower fractal dimension of veins in zone C was the only parameter to be associated with a reduction in mean RNFL thickness (*P* = 0.028). Tham et al. reported a similar finding, as each SD decrease in fractal dimension was associated with a 1.60 μm decrease in average RNFL thickness (*P* = 0.002) [21]. This low branching complexity could be a sign of microvascular alteration and neurodegeneration revealed by the decrease of RNFL thickness in our elderly population. However, in the subgroup analysis (Table 3), the association between a lower fractal dimension of veins in zone C and a reduction in mean RNFL thickness is no more significant. We could make the hypothesis that presence of hypertension or diabetes affect first and foremost retinal fractal dimension.

We also found that a thinner global RNFL was associated with sex, diabetes, and longer eyes. Regarding sex, we found in our population that male subjects had a significantly lower global RNFL thickness than females. The Beijing Eye Study showed women to have a higher RNFL thickness [38]. Mauschitz et al. found that sex was no longer associated with RNFL thickness after correcting for axial length, which was on average shorter in women and

**Table 3. Associations between retinal vascular parameters and global retinal nerve fiber layer thickness in participants without hypertension and diabetes.**

| Retinal vascular parameters (per SD decrease) | Mean ± SD | β (SE)[1] | P-value | β (SE)[2] | P-value | β (SE)[3] | Unajusted P-value[4] | FDR Adjusted P-value[5] |
|---|---|---|---|---|---|---|---|---|
| Caliber, μm | | | | | | | | |
| Six largest arterioles in zone B | 139.73 ± 13.90 | -4.01 (0.70) | < 0.001 | -3.99 (0.71) | < 0.001 | -1.88 (0.78) | 0.0165 | 0.037 |
| Six largest veins in zone B | 193.86 ± 20.20 | -5.53 (0.72) | < 0.001 | -5.54 (0.73) | < 0.001 | -3.28 (0.78) | < 0.001 | < 0.001 |
| Six largest arterioles in zone C | 143.96 ± 13.67 | -3.91 (0.72) | < 0.001 | -3.89 (0.73) | < 0.001 | -2.02 (0.77) | 0.009 | 0.027 |
| Six largest veins in zone C | 201.47 ± 19.92 | -5.33 (0.73) | < 0.001 | -5.32 (0.73) | < 0.001 | -3.27 (0.79) | < 0.001 | < 0.001 |
| Fractal dimension | | | | | | | | |
| Total zone C | 1.39 ± 0.06 | -2.43 (0.80) | 0.005 | -2.46 (0.81) | 0.003 | -1.07 (0.83) | 0.198 | 0.251 |
| Arterioles zone C | 1.19 ± 0.06 | -2.51 (0.76) | 0.001 | -2.54 (0.76) | < 0.001 | -1.21 (0.77) | 0.119 | 0.179 |
| Veins zone C | 1.16 ± 0.05 | -1.78 (0.80) | 0.027 | -1.79 (0.81) | 0.028 | -0.51 (0.79) | 0.523 | 0.523 |
| Vascular Tortuosity | | | | | | | | |
| Simple tortuosity, vessels, × $10^4$ | 1.09 ± 0.02 | -2.04 (1.73) | 0.242 | -1.87 (1.74) | 0.284 | -2.15 (1.75) | 0.223 | 0.251 |
| Curvature tortuosity, vessels, × $10^4$ | 0.73 ± 0.13 | -1.17 (0.88) | 0.186 | -1.11 (0.88) | 0.213 | -1.49 (0.91) | 0.102 | 0.179 |

SD, standard deviation; SE, standard error; FDR, false discovery rate.

[1]Crude associations.

[2]Adjusted for age and sex.

[3]Adjusted for age, sex and axial length.

[4]Unadjusted p-values.

[5]Corrected p-values for multiple testing.

confounded the effect of sex on RNFL thickness [34]. Regarding diabetes, most studies report that patients with diabetes present with thinner RNFL thickness [39, 40]. One explanation could be neurodegeneration to be a consequence of microvascular damages and inflammation described in diabetic patients. A recent longitudinal study showed that diabetes was associated with accelerated RNFL loss regardless of the presence and progression of diabetic retinopathy or the intraocular pressure profile, suggesting that diabetic neurodegeneration may precede the microvascular abnormalities of diabetic retinopathy [41]. In this present study, we did not collect information about the duration of the diabetes and glycemic stability. These parameters should be taken into account in future studies. Regarding the association between greater axial length and thinner RNFL thickness, our findings are in line with several studies [42–45]. The mechanism remains unclear, but could be the result of longer axial length or measurement modifications due to magnification factors [46]. In contrast to Lee et al., in this study we did not compare RNFL thickness between participants with and without hypertension [47]. Nevertheless, hypertension was added to the adjusted-multivariable model. The association between hypertension history and RNFL thickness warranted further studies.

Strengths of our study include its large elderly population-based sample, extensive retinal vascular parameters phenotype (caliber, fractal dimension and vascular tortuosity) and a sub-group analysis in participants without hypertension or diabetes.

Potential limitations of this study should be mentioned. First, retinal vessel diameter has been reported to vary during the cardiac cycle [48–50]. Even though a validation study demonstrated that retinal parameters measured with SIVA are relatively stable, fundus photography analysis involves instantaneous images and cannot take into account vessel caliber variations [51]. Adaptive optics imaging could be an alternative, since this enables dynamic and high-resolution imaging and thereby integrates vessel caliber variations [52]. Second, our findings in an urban Caucasian European population cannot be extrapolated to different ethnic groups.

Third, the causative relationship between retinal vascular geometry and RNFL thickness cannot be definitively assessed owing to the cross-sectional nature of our design. Fourth, we did not take into account the ocular magnification in the SIVA measurements. It could increase variation in RNFL thickness around the 3.5-mm-diameter measurements which may impact the analysis for extreme axial length eyes. Finally, the precision of retinal vasculature grading may be affected by measurement errors related to subjective grader input, variability in image, and contrast or brightness factor errors. There are no clear explanations as to why some parameters in zone B are associated with RNFL but not with zone C parameters. At present, measurement areas are limited owing to the time consumed to track vasculature. Future research using artificial intelligence technologies to aid segmenting vasculature will expand the potential of this analysis.

In conclusion, we found that a narrower retinal vasculature was associated with a thinner global RNFL in eyes without optic neuropathy. The clinical relevance of these findings warrants further studies.

## Supporting information

**S1 Table. Baseline characteristics between participants and non-participants in the Montrachet study.**
(DOCX)

**S2 Table. Associations between retinal vascular parameters and sectorial temporal retinal nerve fiber layer thickness.**
(DOCX)

**S3 Table. Associations between retinal vascular parameters and sectorial nasal retinal nerve fiber layer thickness.**
(DOCX)

## Acknowledgments

The authors thank Sandrine Daniel for her valuable skills in data management for the Montrachet study.

## Author Contributions

**Conceptualization:** Christine Binquet, Catherine Creuzot-Garcher.

**Data curation:** Alassane Seydou, Christine Binquet.

**Formal analysis:** Alassane Seydou, Abderrahmane Bourredjem.

**Methodology:** Abderrahmane Bourredjem.

**Resources:** Ryo Kawasaki.

**Software:** Abderrahmane Bourredjem, Ryo Kawasaki.

**Validation:** Christine Binquet.

**Writing – original draft:** Louis Arnould, Martin Guillemin.

**Writing – review & editing:** Pierre-Henry Gabrielle, Alain M. Bron, Catherine Creuzot-Garcher.

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
