## [Decision Letter · Decision Letter 0]

13 May 2020

PONE-D-20-00420

Association Between the Retinal Vascular Network and Retinal Nerve Fiber Layer in the Elderly: The Montrachet Study

PLOS ONE

Dear Dr Arnould,

Thank you for submitting your manuscript to PLOS ONE. After careful consideration, we feel that it has merit but does not fully meet PLOS ONE’s publication criteria as it currently stands. Therefore, we invite you to submit a revised version of the manuscript that addresses the points raised during the review process.

We would appreciate receiving your revised manuscript by Jun 27 2020 11:59PM. To enhance the reproducibility of your results, we recommend that if applicable you deposit your laboratory protocols in protocols.io, where a protocol can be assigned its own identifier (DOI) such that it can be cited independently in the future. For instructions see: http://journals.plos.org/plosone/s/submission-guidelines#loc-laboratory-protocols

We look forward to receiving your revised manuscript.

Kind regards,

Ireneusz Grulkowski, PhD

Academic Editor

PLOS ONE

3. Thank you for including your ethics statement: "The study was approved by the regional ethics committee. It followed the tenets of the Declaration of Helsinki and it was registered as 2009-A00448-49."

"Supported by an interregional grant (Programme Hospitalier de Recherche Clinique) and the Regional Council of Burgundy; by INRA, CNRS, Université de Bourgogne, Regional Council of Burgundy France (PARI Agrale 1), FEDER (European Funding for Regional Economic Development); and a French Government grant managed by the French National Research Agency (ANR) under the ‘‘Investissements d’Avenir’’ program, ANR-11-LABX-0021-01-LipSTIC Labex."

"No. the funders had no role in study design, data collection and analysis, decision to publish, or preparation of the manuscript"

6. Thank you for stating the following in the Competing Interests section:

"Disclosure: M. Guillemin, None; L. Arnould, None; A. Seydou, None; P H. Gabrielle, None; A. Bourredjem, None; R. Kawasaki, None; C. Binquet, None; A M. Bron, Aerie (C), Allergan (C), Baush and Lomb (C), Santen Pharmaceutical (C), Théa (C); C. Creuzot-Garcher, Allergan (C), Bayer (C), Horus (C), Novartis (C), Roche (C), Théa (C)."

Reviewers' comments:

Reviewer's Responses to Questions

**Comments to the Author**

1. Is the manuscript technically sound, and do the data support the conclusions?

Reviewer #1: Yes

Reviewer #2: Partly

2. Has the statistical analysis been performed appropriately and rigorously? 

Reviewer #1: I Don't Know

Reviewer #2: I Don't Know

3. Have the authors made all data underlying the findings in their manuscript fully available?

Reviewer #1: Yes

Reviewer #2: Yes

4. Is the manuscript presented in an intelligible fashion and written in standard English?

Reviewer #1: Yes

Reviewer #2: Yes

5. Review Comments to the Author

Reviewer #1: In this manuscript, Arnould, et al. report that in elderly study participants without optic neuropathy, thinner retinal nerve fiber layer (RNFL) thickness are associated with reduced caliber of retinal arteries and veins. They also report RNFL thickness is associated with several demographic features including sex, self-declared history of diabetes, and axial length.

Because several authors have published similar results, could the authors highlight or compare and contrast what is novel about their findings? Is it the age of the study participants? Is it the method of analyzing the retinal vessels?

Half of the study participants reported some degree of hypertension. Was there a significant difference in RNFL thickness or retinal vascular metrics for those with hypertension compared to those without?

The purpose of the manuscript defined in the introduction is the detailed analysis comparing retinal vasculature to RNFL thickness. I would recommend emphasizing these findings in the conclusion of the abstract which is vaguely written currently. Also, the discussion should be re-organized to focus on retinal vasculature metrics association with RNFL thickness and then followed by RNFL associations with demographic features.

What was the mean RNFL? How does this compare to other studies?

Could they authors explain their low rate (10 cases out of 1153) of age-related macular degeneration compared to reported prevalence data (Prevalence and incidence of age-related macular degeneration in Europe: a systematic review and meta-analysis. British Journal of Ophthalmology; 11 November 2019)?

Reviewer #2: Arnould et al. present a manuscript assessing the relationship of arterioles and venules to the RNFL thickness measurements. As noted by the authors, many studies have recently been published showing a link between the RNFL and ocular, systemic and neurodegenerative diseases. Their question, of how this layer is related in the elderly and with respect to the vasculature is an important component to this ongoing research.

This manuscript looks specifically at the relationship of different retinal vasculature morphology metrics (on a fundus image) to the thickness of the RNFL at 3.5mm from the optic disc. They find that arteriole and venous vessel caliber is related to RNFL thickness along with age. While this paper is interesting and normative data is needed to understand how aging affects the retina, this author has several concerns that need to be addressed.

Major Issues:

1. The inclusion of subjects with diabetes means that the relationship reported isn’t purely based on age although the authors do adjust for this by using diabetes (and hypertension as well) as a factor in one of their linear regressions. However, nothing is said about the prevalence of retinopathy (which can happen with both diabetes and hypertension). Additionally, there’s also a known relationship between diabetic retinopathy and the length of time someone has diabetes. Yet none of these things are taken into account leading this reviewer to wonder how the regression and these relationships would change. This reviewer would appreciate a deeper examination of subjects with diabetes and a better understanding of that population’s medical information. Additionally, the exclusion criteria does not mention how other ocular diseases were treated in terms of including or excluding those populations (i.e. sickle cell disease, Macular Telangiectasia etc.).

2. Was ocular magnification taken into account when measuring 3.5 mm away from the optic disc? While axial length itself does not impact thickness measurements (PMID: 30025118), it does impact the placement of where these measurements take place if from a set distance. With longer axial lengths, the circle, if ocular magnification is not taken into account, will cover a larger retinal distance than other subjects with short axial lengths. This leads to increased variation in RNFL thickness measurements which may impact the analysis. Please clarify if ocular magnification was corrected for. If not, it would be preferable for the authors to correct for ocular magnification so that 3.5 mm measurements were the same for each individual. If this is not possible, the lack of correction should be noted in the limitations paragraph.

3. Table 1: Stated both in the table and throughout the paper, it says that 970 subjects fit the criteria for the study yet there are additional n values in the same column as the baseline characteristic with no mention of what they mean. If only 878 subjects of the 970 subjects had axial length measurements, would this not also impact your findings for the regression? Given that some of these factors (such as axial length) are known to impact RNFL thickness, why include these subjects? Please clarify what the n values for the table mean. If subjects were included without a full set of data as presented, including them for the regression may be misleading and the authors should consider removing these subjects from their analysis.

4. This reviewer appreciates your transparency with the various regressions that you ran. However, with reading this paper, it is often difficult to remember the central point is to show that even with age, the relationship between retinal vasculature and RNFL thickness holds. If the authors could work to improve the clarity of their writing and work to focus the results in particular, this reviewer thinks it would read significantly easier and the main points would come across much stronger rather than the potential of readers to be caught in the details.

Minor Issues

1. Line 59: The sentence states “using fundus and other techniques”. Please expand on these other techniques and discuss the benefits of the fundus over these other techniques.

2. Lines 64-69: Multiple different software is cited but nothing is stated about why this goes into the introduction. I would expect a reason for why SIVA was used if this is brought up in the introduction over the others, or is the benefit of a fundus that there are these automated techniques?

3. Line 107-109: The sentence about the technicians controlling for image quality and segmentation is ambiguous. Was there an image quality metric with a cut off, were the technicians given any information or guidance or were these metrics purely based on their expertise?

4. Is it appropriate to keep p < 0.05 for all tests given the number of statistical tests done? This reviewer would suggest consulting with a biostatistician (if not already done) to make sure that the analysis provided is accurate and that the p-values do not need adjustment given the number of analyses this paper employs.

6. PLOS authors have the option to publish the peer review history of their article (what does this mean?). If published, this will include your full peer review and any attached files.

Reviewer #1: No

Reviewer #2: No

---

## [Author Response · Author response to Decision Letter 0]

21 Jul 2020

Manuscript ID PONE-D-20-00420

Association Between the Retinal Vascular Network and Retinal Nerve Fiber Layer in the Elderly: The Montrachet Study

Thank you for giving us the opportunity to revise this submission following peer review. After consultation with the other authors, we have responded to the reviewers’ constructive suggestions by revising the manuscript as attached. We would like to take the opportunity to respond to the individual points below.

Reviewer #1: In this manuscript, Arnould, et al. report that in elderly study participants without optic neuropathy, thinner retinal nerve fiber layer (RNFL) thickness are associated with reduced caliber of retinal arteries and veins. They also report RNFL thickness is associated with several demographic features including sex, self-declared history of diabetes, and axial length.

Because several authors have published similar results, could the authors highlight or compare and contrast what is novel about their findings? Is it the age of the study participants? Is it the method of analyzing the retinal vessels?

As suggested, we added these sentences in the revised version of our manuscript as follows (line 265-282), first paragraph of the discussion

“We found that thinner retinal vessels caliber and reduced fractal dimension of veins in zone C were associated with a thinner global RNFL thickness in non-glaucomatous participants. These results are in keeping with other published data. In a cross-sectional investigation of Asian participants, each SD decrease in retinal arteriolar caliber was associated with a 5.81 μm decrease in the mean global measured RNFL thickness (P < 0.001), and each SD decrease in retinal venular caliber was associated with an 8.37 μm decrease in mean global RNFL thickness (P < 0.001).43 In this study, participants’ mean age was 56.8 years. In a study derived from the Singapore Chinese Eye Study, each SD decrease in the retinal arteriolar caliber was associated with a 1.60 μm decrease in global RNFL thickness (P = 0.002), and each SD decrease in venular caliber was associated with a 1.97 μm decrease in average RNFL thickness (P < 0.001) (mean age was 53.6 ± 6.7 years).22 Finally, this association was also confirmed in a middle-aged Caucasian population, where each 10 μm increase in the retinal arteriolar and venular caliber was associated with a 5.58 μm and a 3.79 μm increase in mean global RNFL thickness, respectively (P < 0.001) (mean age was 47.3 ± 0.9 years).21 We confirmed in our study the association between RNFL thickness and retinal vessels caliber in an elderly population.” 

Half of the study participants reported some degree of hypertension. Was there a significant difference in RNFL thickness or retinal vascular metrics for those with hypertension compared to those without?

We agree with the concern raised by the Reviewer. Indeed, Mauschitz et al. investigated pooled data from 8 European population-based studies (including the Montrachet Study) and showed reduced RNFL in hypertensive patients (Mauschitz et al. Ophthalmology 2018). Moreover, we fully agree with the Reviewer’s comment concerning hypertension and retinal vascular metrics. Ponto et al. showed that systemic hypertension was associated with lower values of arteriolar retinal caliber (Ponto et al. J Hypertension 2017). Our primary objective was to study the association between RNFL and retinal vascular parameters independently of cardiovascular risk factors. We did not perform in our analysis a comparison between participants with and without hypertension regarding their RNFL thickness as we decided to use this variable in our adjusted model. We added the following sentence in the revised version of our manuscript (line 343-347).

“In contrast to Lee et al., we did not compare RNFL thickness between participants with and without hypertension.47 Indeed, hypertension is known to be associated with reduced RNFL and decreased arteriolar retinal caliber.34,48 As a consequence, we added hypertension history to the adjusted-multivariable model. The association between hypertension history, RNFL thickness and retinal vascular metrics warrants further studies”

The purpose of the manuscript defined in the introduction is the detailed analysis comparing retinal vasculature to RNFL thickness. I would recommend emphasizing these findings in the conclusion of the abstract which is vaguely written currently. 

We agree with the Reviewer and we modified the conclusion of the abstract as follows (line 46-47).

“In this study, we found that decreased retinal vessels caliber was associated with a decreased RNFL thickness in the elderly without optic neuropathy.”

Also, the discussion should be re-organized to focus on retinal vasculature metrics association with RNFL thickness and then followed by RNFL associations with demographic features.

We thank the Reviewer for emphasizing this crucial point. We re-organized the discussion as mentioned (line 265-282) and (line 324-347). 

What was the mean RNFL? How does this compare to other studies?

The mean RNFL of the study participants was 92.08 ± 13.29 μm. This result was similar to other population-based studies. The mean RNFL thickness in the Beijing Eye Study was 103.2 ± 12.6 µm (66.2 ± 9.9 years) (YA Xing Wang et al. PlosOne 2013). The mean RNFL thickness in the Alienor study was 86.8 ± 13.7 µm (81.6 ± 4.2 years) (Juan Luis Méndez-Gómez et al. JAMA Netw Open 2018). We added the following sentence in the revised version of our manuscript (line 196)

“The mean RNFL of the study participants was 92.08 ± 13.29 μm”

Could they authors explain their low rate (10 cases out of 1153) of age-related macular degeneration compared to reported prevalence data (Prevalence and incidence of age-related macular degeneration in Europe: a systematic review and meta-analysis. British Journal of Ophthalmology; 11 November 2019)?

We thank the Reviewer for highlighting this point. Prevalence of age-related macular degeneration (AMD) in the Montrachet study was reported (Creuzot-Garcher et al. Acta Ophthalmologica 2015). The prevalence is lower compared to reported prevalence data because we investigated a white, urban, and generally healthy population with high economic status. Moreover, in this study, we only excluded participants with late AMD and modification of the vascular architecture in zone B and C attested by a trained grader. As suggested by the Reviewer, we added the following statement in the revised version of our manuscript (line 162-164) and we modified the caption of the Figure.

“Patients presenting with late age-related macular degeneration (AMD) and modification of the vascular architecture in zone B and C, attested by a trained grader, were not retained for the analysis.”

Reviewer #2: Arnould et al. present a manuscript assessing the relationship of arterioles and venules to the RNFL thickness measurements. As noted by the authors, many studies have recently been published showing a link between the RNFL and ocular, systemic and neurodegenerative diseases. Their question, of how this layer is related in the elderly and with respect to the vasculature is an important component to this ongoing research.

This manuscript looks specifically at the relationship of different retinal vasculature morphology metrics (on a fundus image) to the thickness of the RNFL at 3.5mm from the optic disc. They find that arteriole and venous vessel caliber is related to RNFL thickness along with age. While this paper is interesting and normative data is needed to understand how aging affects the retina, this author has several concerns that need to be addressed.

Major Issues:

1. The inclusion of subjects with diabetes means that the relationship reported isn’t purely based on age although the authors do adjust for this by using diabetes (and hypertension as well) as a factor in one of their linear regressions. However, nothing is said about the prevalence of retinopathy (which can happen with both diabetes and hypertension). Additionally, there’s also a known relationship between diabetic retinopathy and the length of time someone has diabetes. Yet none of these things are taken into account leading this reviewer to wonder how the regression and these relationships would change. This reviewer would appreciate a deeper examination of subjects with diabetes and a better understanding of that population’s medical information. 

We fully understand this comment. As reported in our previous study (Creuzot-Garcher et al. Acta Ophthalmologica 2015), there were 9 participants of the Montrachet study with an history of diabetic retinopathy. In our study, we found in our database that only 2 diabetic patients presented with diabetic retinopathy. We decided not to exclude these patients because diabetes was chosen to be an adjustment variable in the linear regression. The primary objective of our study was to investigate the association between retinal nerve fiber layer thickness and retinal vascular architectural parameters independently of cardiovascular risk factors. Unfortunately, we did not collect information about the duration of the diabetes and the glycemic status. However, participants in the Montrachet study were healthy volunteers and their cardiovascular risk factors were well monitored. 

We added the following statement in the revised version of our manuscript.

Line 195-196 

“Two participants with diabetes and diabetic retinopathy were kept in the analysis.”

Line 337-338 

“In the present study, we did not collect information about the duration of the diabetes and glycemic stability. These parameters should be taken into account in future studies.”

Additionally, the exclusion criteria does not mention how other ocular diseases were treated in terms of including or excluding those populations (i.e. sickle cell disease, Macular Telangiectasia etc.).

Participants ophthalmic history were collected in the same way for the entire population of the Montrachet study. In previous studies focusing on retinal history in the Montrachet study (Ben Ghezala et al. Retina 2020 and Gabrielle et al. IOVS 2019), we did not find other ocular vascular diseases. 

2. Was ocular magnification taken into account when measuring 3.5 mm away from the optic disc? While axial length itself does not impact thickness measurements (PMID: 30025118), it does impact the placement of where these measurements take place if from a set distance. With longer axial lengths, the circle, if ocular magnification is not taken into account, will cover a larger retinal distance than other subjects with short axial lengths. This leads to increased variation in RNFL thickness measurements which may impact the analysis. Please clarify if ocular magnification was corrected for. If not, it would be preferable for the authors to correct for ocular magnification so that 3.5 mm measurements were the same for each individual. If this is not possible, the lack of correction should be noted in the limitations paragraph.

We agree with the concerns raised by the Reviewer. We added the following statement in the revised version of our manuscript (line 377-380).

“Fourth, we did not take into account the ocular magnification in the SIVA measurements. It could increase variation in RNFL thickness around the 3.5-mm-diameter measurements which may impact the analysis for extreme axial length eyes.”

3. Table 1: Stated both in the table and throughout the paper, it says that 970 subjects fit the criteria for the study yet there are additional n values in the same column as the baseline characteristic with no mention of what they mean.

We thank the Reviewer for highlighting this point. The n values in the first column of Table 1 and Table 2 correspond to the available data for each variable. We agree with the Reviewer that additional n values in the same column with no clear explanation is confusing. Therefore, we placed the n values in the notes under Table 1 and Table 2 to avoid any confusion.

If only 878 subjects of the 970 subjects had axial length measurements, would this not also impact your findings for the regression? 

We fully agree with Reviewer’s comment. Missing data represent less than 10% of our dataset. As a consequence, we did not perform a multiple imputation method. However, it could affect the regression results and it could reduce the statistical power of our associations. 

Given that some of these factors (such as axial length) are known to impact RNFL thickness, why include these subjects?

Axial length is a known confounder factor between all retinal vascular parameters and also for RNFL thickness. In this study, we kept the axial length as an adjustment variable in order to investigate the relationship between retinal vascular parameters and RNFL thickness irrespectively of the ametropia. 

Please clarify what the n values for the table mean. If subjects were included without a full set of data as presented, including them for the regression may be misleading and the authors should consider removing these subjects from their analysis.

As suggested by the Reviewer we removed the participants with missing axial length variable in the multivariable models and a note has been written under the Tables in the revised version of our manuscript.

4. This reviewer appreciates your transparency with the various regressions that you ran. However, with reading this paper, it is often difficult to remember the central point is to show that even with age, the relationship between retinal vasculature and RNFL thickness holds. If the authors could work to improve the clarity of their writing and work to focus the results in particular, this reviewer thinks it would read significantly easier and the main points would come across much stronger rather than the potential of readers to be caught in the details.

Thank you to the Reviewer for emphasizing this crucial point. We re-organized the discussion as mentioned. (line 265-282) and (line 324-347).

Minor Issues

1. Line 59: The sentence states “using fundus and other techniques”. Please expand on these other techniques and discuss the benefits of the fundus over these other techniques.

We thank the Reviewer for highlighting this point. It is possible to investigate the human microcirculation network with noninvasive devices such as fundus photographs, capillaroscopy, renal doppler or brain imaging. In the Montrachet study, participants benefited from an exhaustive eye examination. Fundus photographs were performed for each participant. Compared to the other techniques, fundus photographs are less expensive, and it can be performed by a technician or a nurse. Moreover, semi-automated software based on fundus are available to thoroughly describe the retinal vascular network with quantitative metrics. Finally, it has been shown that retinal vascular network is reliable to assess the systemic vascular status (Arnould et al. PLoS One 2018). We deleted “other techniques” in the revised manuscript (line 65). 

2. Lines 64-69: Multiple different software is cited but nothing is stated about why this goes into the introduction. I would expect a reason for why SIVA was used if this is brought up in the introduction over the others, or is the benefit of a fundus that there are these automated techniques?

We fully agree with the Reviewer’s comment. We used SIVA software because at the time of the data collection in the Montrachet study (2009-2013), it was the only software that was able to quantify complex vascular parameters such as fractal dimension and tortuosity. Other softwares were only focused on retinal caliber. We deleted information about other softwares in the introduction of the revised manuscript (line 62-70).

3. Line 107-109: The sentence about the technicians controlling for image quality and segmentation is ambiguous. Was there an image quality metric with a cut off, were the technicians given any information or guidance or were these metrics purely based on their expertise?

We thank the reviewer for highlighting this point. We modified the following statement in the revised version of our manuscript (line 121-122).

“Signal strength and the quality of segmentation were controlled by technicians based on their expertise and training”

4. Is it appropriate to keep p < 0.05 for all tests given the number of statistical tests done? This reviewer would suggest consulting with a biostatistician (if not already done) to make sure that the analysis provided is accurate and that the p-values do not need adjustment given the number of analyses this paper employs.

We agree with the Reviewer that, given the number of statistical tests in this study, keeping p < 0.05 for all tests is not appropriate. As suggested by the Reviewer, we used the procedure described by Benjamini and Hochberg (False Discovery Rate correction) (Benjamin Y et al . J R Stat Soc. 1995) to correct p values for multiple testing. FDR Adjusted p values are presented in multivariable models for each independent variable in Tables 3,4 and 5 in the revised version of our manuscript. Significant associations remained stable with this correction except the associations between fractal dimension (total zone C and arterioles zone C) and nasal RNFL thickness. We added the following statement in the revised version of our manuscript (line 183-185).

“Given the number of statistical tests in this study, we used the procedure of False Discovery Rate (FRD) correction to correct p values for multiple testing.31”

---

## [Decision Letter · Decision Letter 1]

21 Aug 2020

PONE-D-20-00420R1

Association Between the Retinal Vascular Network and Retinal Nerve Fiber Layer in the Elderly: The Montrachet Study

PLOS ONE

Dear Dr. Arnould,

Thank you for submitting your manuscript to PLOS ONE. After careful consideration, we feel that it has merit but does not fully meet PLOS ONE’s publication criteria as it currently stands. Therefore, we invite you to submit a revised version of the manuscript that addresses the points raised during the review process.

We look forward to receiving your revised manuscript.

Kind regards,

Ireneusz Grulkowski, PhD

Academic Editor

PLOS ONE

Additional Editor Comments (if provided):

Please, analyze the reviewers' comments and revise the paper again.

Reviewers' comments:

Reviewer's Responses to Questions

**Comments to the Author**

1. If the authors have adequately addressed your comments raised in a previous round of review and you feel that this manuscript is now acceptable for publication, you may indicate that here to bypass the “Comments to the Author” section, enter your conflict of interest statement in the “Confidential to Editor” section, and submit your "Accept" recommendation.

Reviewer #1: (No Response)

Reviewer #2: (No Response)

2. Is the manuscript technically sound, and do the data support the conclusions?

Reviewer #1: Partly

Reviewer #2: Yes

3. Has the statistical analysis been performed appropriately and rigorously? 

Reviewer #1: I Don't Know

Reviewer #2: Yes

4. Have the authors made all data underlying the findings in their manuscript fully available?

Reviewer #1: No

Reviewer #2: Yes

5. Is the manuscript presented in an intelligible fashion and written in standard English?

Reviewer #1: (No Response)

Reviewer #2: Yes

6. Review Comments to the Author

Reviewer #1: Thank you for considering revisions. These analyses are difficult in elderly patients because of the confounding variables. For example, in this study nearly 70% of participants have a history of hypertension. Presence of hypertension, and diabetes, likely affects affect retinal vascular parameters, but this information is not presented in the manuscript.

Table 2 shows the association between RNFL to demographics, lifestyle and clinical characteristics. It would be useful to show associations between retinal vascular parameters to demographics, lifestyle and clinical characteristics as well. I would expect there to be significant associations with hypertension, diabetes and possibly axial length. Increased transparency with the multiple variables involved will help readers understand the complexity associated with these analyses.

Could the authors run a separate analysis evaluating for association of RNFL and retinal vasculature parameters after excluding hypertension, diabetes and extreme axial length differences? It is unclear if the numbers would support this evaluation.

For the RNFL analysis, this work is contrasted by others because it is an “elderly cohort” – could the authors clearly state the mean age in the results section?

Is Table 1 necessary? This is essentially comparing cases that were excluded to participants that were included. Why is this relevant to the comparison of RNFL to retinal vascular parameters? If important to include, perhaps these findings could be summarized in the text of the results.

I would exclude the sentence in the abstract describing thinner global RNFL thickness associated with males, patients with diabetes and longer eyes because these findings are not directly relevant to the purpose of the manuscript.

Why was axial length not included in Table 2? Axial length may confound other associations – such as gender differences.

Reviewer #2: In general, the authors have made considerable steps in improving their manuscript. The paper now reads clearer and the main points are not lost. However, there are still several points of concern about this study for this reviewer.

Major Concerns:

1. As previously mentioned by reviewer 1, several authors have published similar studies. While the discussion now clearly compares the various studies that have been published, it still remains unclear what is novel about this study or its contribution to the literature. Please highlight these points in your discussion.

2. The exclusion criteria state that subjects with epiretinal membranes and late-stage AMD were excluded due to known alterations these pathologies have on the vasculature architecture in the retina. Yet the same argument can be made for subjects with hypertension or with diabetes. Based on this, was the exclusions of the subjects based on the fact that those with ERMs or AMD had primarily ocular diseases and therefore the changes in the retinal vasculature had nothing to do with the cardio-vasculature system or was it truly because the vasculature in the retina is altered? If it’s the former, please update the section of the manuscript. If it’s the latter, then the argument for excluding those subjects but including subjects with diabetes and hypertension needs to be re-examined.

7. PLOS authors have the option to publish the peer review history of their article (what does this mean?). If published, this will include your full peer review and any attached files.

Reviewer #1: No

Reviewer #2: No

---

## [Author Response · Author response to Decision Letter 1]

22 Sep 2020

Manuscript ID PONE-D-20-00420R1

Association Between the Retinal Vascular Network and Retinal Nerve Fiber Layer in the Elderly: The Montrachet Study

Thank you for giving us the opportunity to revise this submission following peer review. After consultation with the other authors, we have responded to the reviewers’ constructive suggestions by revising the manuscript as attached. We 

would like to take the opportunity to respond to the individual points below.

Reviewer #1: Thank you for considering revisions. These analyses are difficult in elderly patients because of the confounding variables. For example, in this study nearly 70% of participants have a history of hypertension. Presence of hypertension, and diabetes, likely affects affect retinal vascular parameters, but this information is not presented in the manuscript.

Table 2 shows the association between RNFL to demographics, lifestyle and clinical characteristics. It would be useful to show associations between retinal vascular parameters to demographics, lifestyle and clinical characteristics as well. I would expect there to be significant associations with hypertension, diabetes and possibly axial length. Increased transparency with the multiple variables involved will help readers understand the complexity associated with these analyses.

We thank the Reviewer for highlighting this point. We previously presented associations between retinal vascular parameters and demographics, lifestyle and clinical characteristics in our manuscript : Association between the retinal vascular network with Singapore “I” Vessel Assessment (SIVA) software, cardiovascular history and risk factors in the elderly: The Montrachet study. (Arnould L et al. PlosOne April 2018). As suggested by the Reviewer, we found significant associations between retinal vascular parameters and hypoglycemic treatment, history of systemic vascular events and increased Heart Score risk. For greater clarity, we did not repeat these results. 

Could the authors run a separate analysis evaluating for association of RNFL and retinal vasculature parameters after excluding hypertension, diabetes and extreme axial length differences? It is unclear if the numbers would support this evaluation.

Thank you to the Reviewer for emphasizing this crucial point. We ran a separate analysis in participants without hypertension and diabetes. This analysis is presented in Table 3. We added these sentences in the revised version of our manuscript as follows

Line 220-225

“In participants without hypertension and diabetes (n = 294), after adjustment for age, sex and axial length, each SD decrease in the caliber of the six largest arterioles and veins in zone B and in the caliber of the six largest arterioles and veins in zone C was associated with a decrease in global RNFL thickness (β = −1.88 μm, P = 0.0371; β = −3.28 μm, P < 0.001; β = −2.02 μm, P < 0.001; and β = −3.27 μm, P < 0.001, respectively) (Table 3).”

Line 271-272

“Moreover, in a subgroup analysis (participants without hypertension or diabetes), we found similar results.”

Line 306-310

“However, in the subgroup analysis (Table 3), the association between a lower fractal dimension of veins in zone C and a reduction in mean RNFL thickness is no more significant. We could make the hypothesis that presence of hypertension or diabetes affect first and foremost retinal fractal dimension.” 

For the RNFL analysis, this work is contrasted by others because it is an “elderly cohort” – could the authors clearly state the mean age in the results section?

The mean age in this study was 82.06 (± 3.64) years old. We added this information in the results section as follows (Line 184)

“The mean RNFL of the study participants was 92.08 ± 13.29 μm and the mean age was 82.06 ± 3.64 years old.”

Is Table 1 necessary? This is essentially comparing cases that were excluded to participants that were included. Why is this relevant to the comparison of RNFL to retinal vascular parameters? If important to include, perhaps these findings could be summarized in the text of the results.

We agree with the concern raised by the Reviewer. Table 1 is now part of the supporting information. In the revised version of our manuscript, it is reported as S1 Table. We added these sentences in the revised version of our manuscript as follows (Line 180-181)

“The characteristics of the study population and a comparison between participants and non-participants are summarized in the supporting information (S1 Table).”

Moreover, we transferred Table 4 and 5 in the supporting information (S2 and S3 Table). We added a caption section as requested in the revised version of our manuscript.

I would exclude the sentence in the abstract describing thinner global RNFL thickness associated with males, patients with diabetes and longer eyes because these findings are not directly relevant to the purpose of the manuscript.

We fully agree with the reviewer’s comment. We deleted this sentence in the abstract of the revised manuscript.

Why was axial length not included in Table 2? Axial length may confound other associations – such as gender differences.

We agree with the concerns raised by the Reviewer. We added this information in the revised version of our manuscript (Table 1).

Reviewer #2: In general, the authors have made considerable steps in improving their manuscript. The paper now reads clearer and the main points are not lost. However, there are still several points of concern about this study for this reviewer.

Major Concerns:

1. As previously mentioned by reviewer 1, several authors have published similar studies. While the discussion now clearly compares the various studies that have been published, it still remains unclear what is novel about this study or its contribution to the literature. Please highlight these points in your discussion.

We thank the reviewer for highlighting this point. We added this sentence in the discussion of the revised manuscript (Line 334-337).

“Strengths of our study include its large elderly population-based sample, extensive retinal vascular parameters phenotype (caliber, fractal dimension and vascular tortuosity) and a subgroup analysis in participants without hypertension or diabetes.”

2. The exclusion criteria state that subjects with epiretinal membranes and late-stage AMD were excluded due to known alterations these pathologies have on the vasculature architecture in the retina. Yet the same argument can be made for subjects with hypertension or with diabetes. Based on this, was the exclusions of the subjects based on the fact that those with ERMs or AMD had primarily ocular diseases and therefore the changes in the retinal vasculature had nothing to do with the cardio-vasculature system or was it truly because the vasculature in the retina is altered? If it’s the former, please update the section of the manuscript. If it’s the latter, then the argument for excluding those subjects but including subjects with diabetes and hypertension needs to be re-examined.

We agree with the concerns raised by the Reviewer. We performed a subgroup analysis in participants without hypertension or diabetes (Table 3) and we added the following statement in the revised version of our manuscript (Line 149-152).

“Participants with epiretinal membrane and AMD with modification of the vascular architecture were excluded because they present primary modifications of the retinal vascular profile without systemic involvement.”

---

## [Decision Letter · Decision Letter 2]

8 Oct 2020

Association Between the Retinal Vascular Network and Retinal Nerve Fiber Layer in the Elderly: The Montrachet Study

PONE-D-20-00420R2

Dear Dr. Arnould,

We’re pleased to inform you that your manuscript has been judged scientifically suitable for publication and will be formally accepted for publication once it meets all outstanding technical requirements.

Kind regards,

Ireneusz Grulkowski, PhD

Academic Editor

PLOS ONE

Additional Editor Comments (optional):

Reviewers' comments:

Reviewer's Responses to Questions

**Comments to the Author**

1. If the authors have adequately addressed your comments raised in a previous round of review and you feel that this manuscript is now acceptable for publication, you may indicate that here to bypass the “Comments to the Author” section, enter your conflict of interest statement in the “Confidential to Editor” section, and submit your "Accept" recommendation.

Reviewer #1: All comments have been addressed

Reviewer #2: All comments have been addressed

2. Is the manuscript technically sound, and do the data support the conclusions?

Reviewer #1: Yes

Reviewer #2: (No Response)

3. Has the statistical analysis been performed appropriately and rigorously? 

Reviewer #1: Yes

Reviewer #2: (No Response)

4. Have the authors made all data underlying the findings in their manuscript fully available?

Reviewer #1: Yes

Reviewer #2: (No Response)

5. Is the manuscript presented in an intelligible fashion and written in standard English?

Reviewer #1: Yes

Reviewer #2: (No Response)

6. Review Comments to the Author

Reviewer #1: I would like to thank the authors for making the recommended changes to the manuscript over the past two revisions. Nice work.

Reviewer #2: (No Response)

7. PLOS authors have the option to publish the peer review history of their article (what does this mean?). If published, this will include your full peer review and any attached files.

Reviewer #1: No

Reviewer #2: No

---

## [Editor Report · Acceptance letter]

12 Oct 2020

PONE-D-20-00420R2 

Association Between the Retinal Vascular Network and Retinal Nerve Fiber Layer in the Elderly: The Montrachet Study 

Dear Dr. Arnould:

I'm pleased to inform you that your manuscript has been deemed suitable for publication in PLOS ONE. Congratulations! Your manuscript is now with our production department. 

Kind regards, 

on behalf of

Dr. Ireneusz Grulkowski 

Academic Editor

PLOS ONE